# Identifying host regulators and inhibitors of liver stage malaria infection using kinase activity profiles

Nadia Arang[1,5], Heather S. Kain[1], Elizabeth K. Glennon[1], Thomas Bello[2,3], Denali R. Dudgeon[1], Emily N.F. Walter[1], Taranjit S. Gujral[2,3] & Alexis Kaushansky[1,4]

*Plasmodium* parasites have extensive needs from their host hepatocytes during the obligate liver stage of infection, yet there remains sparse knowledge of specific host regulators. Here we assess 34 host-targeted kinase inhibitors for their capacity to eliminate *Plasmodium yoelii*-infected hepatocytes. Using pre-existing activity profiles of each inhibitor, we generate a predictive computational model that identifies host kinases, which facilitate *Plasmodium yoelii* liver stage infection. We predict 47 kinases, including novel and previously described kinases that impact infection. The impact of a subset of kinases is experimentally validated, including Receptor Tyrosine Kinases, members of the MAP Kinase cascade, and WEE1. Our approach also predicts host-targeted kinase inhibitors of infection, including compounds already used in humans. Three of these compounds, VX-680, Roscovitine and Sunitinib, each eliminate >85% of infection. Our approach is well-suited to uncover key host determinants of infection in difficult model systems, including field-isolated parasites and/or emerging pathogens.

[1] Center for Infectious Disease Research, formerly Seattle Biomedical Research Institute, 307 Westlake Ave N #500, Seattle, WA 98109, USA. [2] Division of Human Biology, Fred Hutchinson Cancer Research Center, 1100 Fairview Ave N, Seattle, WA 98109, USA. [3] Program in Molecular and Cellular Biology, 1959 NE Pacific Street, HSB T-466 University Of Washington Box 357275, Seattle, WA 98195, USA. [4] Department of Global Health, University of Washington, Harris Hydraulics Laboratory Box 357965, Seattle, WA 98195-7965, USA. [5]Present address: Department of Biomedical Sciences, University of California, San Diego, 9500 Gilman Drive, La Jolla, CA 92093, USA. Heather S. Kain, Elizabeth K. Glennon and Thomas Bello contributed equally to this work. Correspondence and requests for materials should be addressed to A.K. (email: alexis.kaushansky@cidresearch.org)

Plasmodium parasites, the causative agents of malaria, inflict tremendous mortality and morbidity worldwide. Yet, the details of how these parasites interact with their host remain largely unexplored. This is particularly true during the first several days of mammalian infection, when the parasite infects and then resides within a single hepatocyte in the liver. While clinically asymptomatic, the liver stage (LS) of development is obligatory for malaria life cycle progression and also houses hypnozoite forms, which are the origin of relapsing malaria[1–4]. The LS is a prime target for therapeutic intervention because parasites are present in very small numbers compared to other stages of infection, which are the typical targets of anti-malarial drugs[5]. The identification of essential host-parasite interactions during this stage could serve as critical points of intervention in preventing blood stage infection and subsequent transmission to mosquitoes.

As for many difficult-to-culture pathogens, systems-level approaches are often incompatible with the challenges of studying LS infection. More robust but experimentally straightforward approaches to identifying host-pathogen interactions and the ability to translate these insights to intervention are desperately needed. Transcriptomic, proteomic and other global approaches have made major strides in recent years (reviewed in ref. [6]). Moreover, new technologies such as genome-wide screens can facilitate the identification of host factors[7–9] involved in the infection. However, many of these approaches require large numbers of infected cells, which are difficult to generate in laboratory strains of malaria and virtually impossible to obtain when pathogens are isolated from the field or other medically relevant settings. Consequently, nearly all drug and vaccine discovery efforts have been performed in laboratory strains of the malaria parasite. Unfortunately, recent studies have demonstrated that these platforms do not always succeed in predicting efficacy in the field due to differences between field and lab strains[10].

Recent work has partially overcome these hurdles and provided initial insights into host factors that mediate LS infection. These findings indicate a role for post-translational regulation of host factors involved in infection[11]. In addition to the direct assessment of post-translational modifications, the transcriptome of infected hepatocytes suggests changes in signal transduction cascades and stress responses, which are classically associated with kinase driven phosphosignaling[12]. Phosphorylation of host molecules Akt, p53 and Bcl-2[11] occurs during LS infection. At least some of these changes, including alterations of the p53[11,13] and Bcl-2 pathways[11,13], have been demonstrated to play a functional role in infection. Inhibition of host cell mitogen-activated protein kinase kinases (MAPKKs) in both Plasmodium-infected erythrocytes and hepatocytes can also curtail infection[14]. Despite these insights, the host kinases that regulate these and other required phosphorylation events during Plasmodium infection remain unknown.

There is also evidence suggesting that protein phosphorylation might directly mediate specific protein-protein interactions between the parasite and the host. Proteins localized to the parasitophorous vacuole membrane (PVM) are modified by phosphorylation in liver[15] and blood[16] stages, although the extent of this post-translational regulatory event remains almost entirely unexplored. While intriguing, existing studies have fallen short of providing a systematic approach to identifying key phosphorylation regulatory networks that govern the development of Plasmodium LS infection. Methodology compatible with the technical challenges of studying LS malaria that can

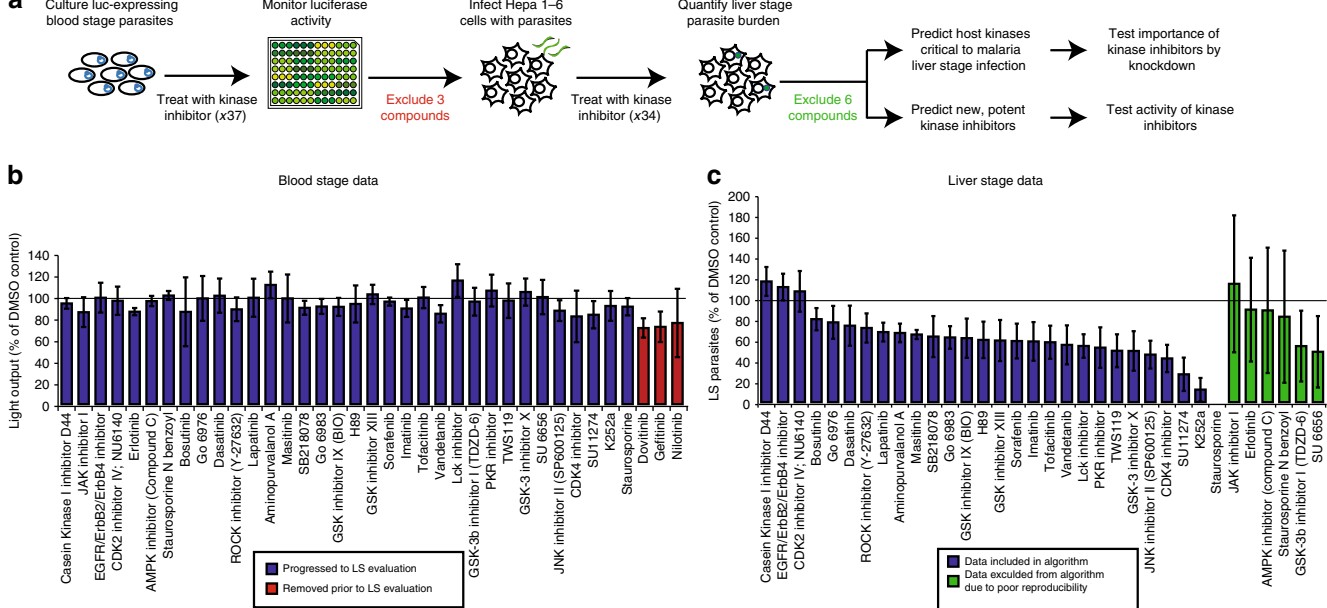

**Fig. 1** Plasmodium LS development is differentially impacted by host-targeted kinase inhibitors. **a** Schematic representing work flow to identify host kinases involved in LS infection by kinase profiling and elastic net regression. **b** P. falciparum GFP-Luciferase expressing blood stage parasites were cultured at 2% parasitemia in 5% hematocrit and evaluated for growth in response to 37 kinase inhibitors at 44 h.p.i. Light output was used as a surrogate measurement for parasite biomass. Values are normalized to the light output of non-treated parasites, which is indicated by a solid line. Kinase inhibitors that exhibited toxicity against blood stage parasites were removed for subsequent study (depicted in red). Data is the average of three independent experiments. **c** 150,000 Hepa 1–6 cells were infected with 50,000 P. yoelii parasites and then treated with kinase inhibitors at 500 nM 1.5 h.p.i. P. yoelii LS development in the presence of kinase inhibitors was evaluated by microscopy at 24 h.p.i. All values are normalized to vehicle-treated control (indicated by a solid line). Kinase inhibitors that exhibited high variability in parasite clearance were excluded from downstream analysis (depicted in green). Data shown is the average of 3–4 independent experiments. Error bars represent standard deviation of independent experiments. Individual data points are provided in Supplementary Table 1

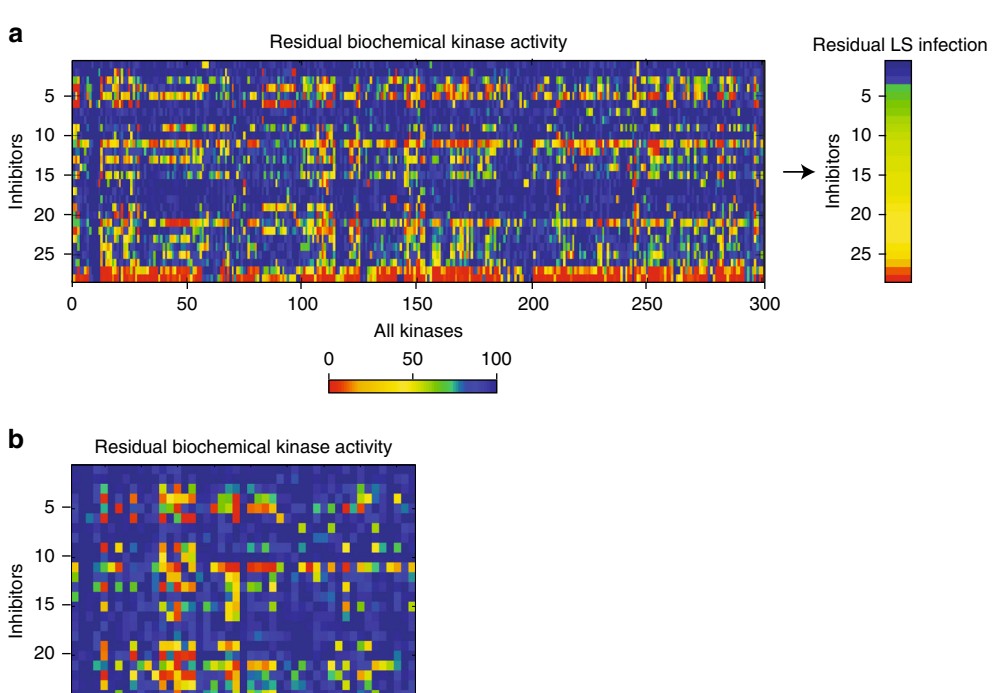

**Fig. 2** Kinase profiles facilitate predictions of key host kinases for LS infection. **a** Heat map depicting residual kinase activity for 300 kinases in response to the 28 kinase inhibitors that were used to train an elastic net regression model. Inhibitors are ranked by their capacity to inhibit LS infection[21]. **b** Heat map depicting residual kinase activity of 47 hit kinases identified in response to selected panel of kinase inhibitors. Inhibitors are ranked by their capacity to inhibit LS infection

systematically identify key host regulatory phosphosignaling networks is needed. Furthermore, methodology that links these host factors to potent inhibitors of infection is required. Towards this end, we have applied an approach that simultaneously predicts key host kinase regulators of *Plasmodium yoelii* LS infection and host-targeted drugs that can eliminate parasite burden in the liver.

Recent approaches have used the combination of experimental data and machine learning algorithms to identify key kinase regulators of a given phenotype[17–19]. For example, Kinase Regression (KiR) aims to identify the kinases that most significantly contribute to a biological phenotype by integrating a straightforward, small-scale kinase inhibitor screen with the tools of computational biology[17]. This approach takes advantage of measurements of polypharmacology, the property of kinase inhibitors to have multiple targets, to identify key kinase regulators of the cellular phenotype of interest. KiR has previously been used in purely mammalian systems, including the identification of novel kinases that regulate cell migration[17] and subsequent metastasis[20]. Related approaches have been used to identify regulators of angiogenesis and proliferation[18,19]. KiR utilizes pre-existing, in vitro activity profiles of 300 commonly studied kinases in response to a collection of 178 kinase inhibitors, including FDA-approved drugs[21]. A small subset of these kinase inhibitors are administered and quantitative phenotypic data is collected.

Here, we take a similar computational approach and integrate experimental phenotypic data with kinase activity profiles by using elastic net regularized regression to predict the most significant host kinase regulators of LS malaria infection as well as

the most inhibitory compounds. A subset of these predictions are then experimentally tested. Taken together, we identify key biological regulators of *Plasmodium yoelii* LS infection and translate these insights into potent host-targeted intervention strategies.

## Results

**Host-targeted inhibitors impact *Plasmodium* LS development.** Here, we adapt an approach that combines experimental data with a machine learning algorithm to identify host kinase regulators of *Plasmodium yoelii* LS infection (Fig. 1a). Thirty-seven kinase inhibitors have previously been described to capture as much of the variability in the inhibitor-kinase activity space as possible using only a modest number of inhibitors[17] (Supplementary Table 1). In this case, the chosen drugs account for >80% variability in the measured kinase activities[17,21]. *Plasmodium*, like its mammalian host, has a diverse repertoire of kinases. Since this approach only incorporates kinase inhibition data on mammalian kinases, any activity that kinase inhibitors have directly on *Plasmodium* kinases has the potential to mislead the computational algorithm. As a surrogate for activity against *Plasmodium* kinases, we evaluated the efficacy of each kinase inhibitor in a *Plasmodium* blood stage growth assay (Fig. 1b)[22]. We reasoned that any inhibitor that cleared the parasites during the asexual blood stage might exert its activity by inhibiting *Plasmodium* kinases, and thus might confound our modeling approach. Three out of the 37 compounds tested exhibited >25% inhibitory activity against asexual blood stage parasite growth compared to non-treated control. To minimize the confounding effect of the inhibition of parasite kinases, these inhibitors were

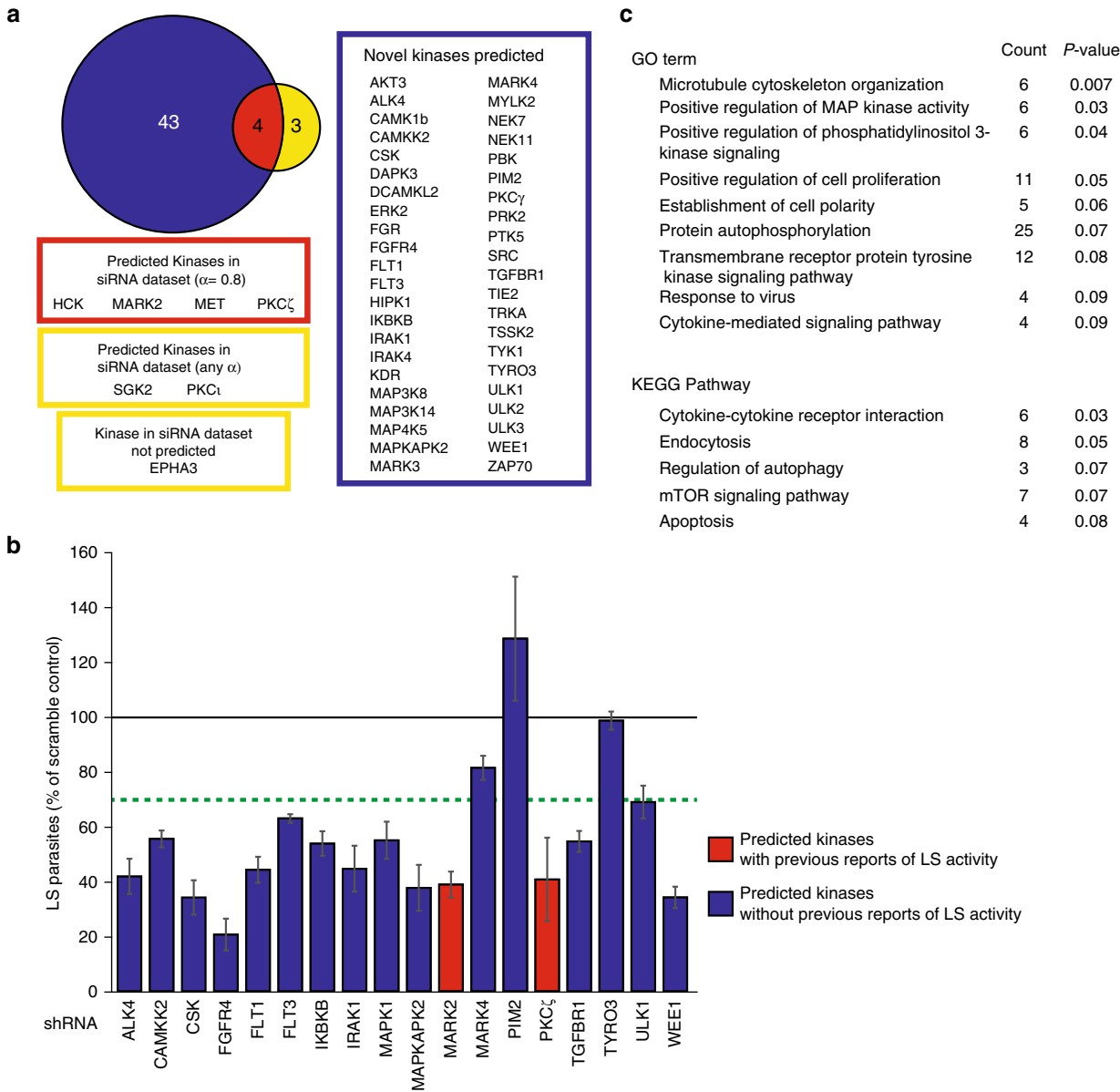

**Fig. 3** Kinase profiling and elastic net regression successfully predict known and novel host kinases important for regulating *P. yoelii* LS infection. **a** Venn diagram depicting overlap between predicted kinase hits and kinases previously reported by whole-kinome siRNA screen. Our approach predicted 47 host kinases to be regulators of *Plasmodium* LS infection. Kinases without any previously described role in LS infection are depicted in blue. Kinases previously implicated in LS infection, and also predicted by the elastic net regression at $\alpha = 0.8$, are depicted in red. Of the kinases previously demonstrated to regulate *Plasmodium* LS infection, those which are predicted at $\alpha < 0.8$ are depicted in yellow. Kinases not recapitulated by our approach at any value of $\alpha$ are shown below. Statistical comparison to existing data was performed using hypergeometric probability test ($p = 0.01$). **b** Bar graph depicting LS development in cells with shRNA-mediated knockdown of a subset of predicted kinases. Values are normalized to non-treated parasites which are indicated by solid line. Green dashed line represents LS burden ≤70% of control. Predicted kinases with previous reports of LS activity are depicted in red. Novel predicted kinases are depicted in blue. Data is representative of at least three independent experiments. Error bars represent standard deviation of technical replicates. **c** Functional enrichment analysis of predicted kinases. The 47 predicted host kinases were analyzed using DAVID Bioinformatics Resources 6.8[25,26]. The 300 kinases that are evaluated by this approach were set as the background for the analysis. *P*-values were determined by modified Fisher's exact test

eliminated from further use in our study (Fig. 1a, b). This does not exclude the possibility that erythrocyte kinases play an important role in *Plasmodium* infection, as has been highlighted in other studies[14,23].

*Plasmodium* LS development can be quantitatively assessed by fluorescence microscopy in Hepa 1–6 mouse hepatoma cells. We infected 150,000 Hepa 1–6 cells with 50,000 *P. yoelii* sporozoites, then administered kinase inhibitors 1.5 hours

post infection (h.p.i.). We assessed LS parasite burden at 24 h.p.i. (Fig. 1a, c). Kinase inhibitors that performed inconsistently over multiple independent experiments, defined as a standard deviation greater than 20% of the mean, were excluded from further analysis (Fig. 1a, c). Remaining kinase inhibitors represent a panel of compounds that differentially and consistently impact parasite burden (Fig. 1c, Supplementary Fig. 1, Supplementary Table 2).

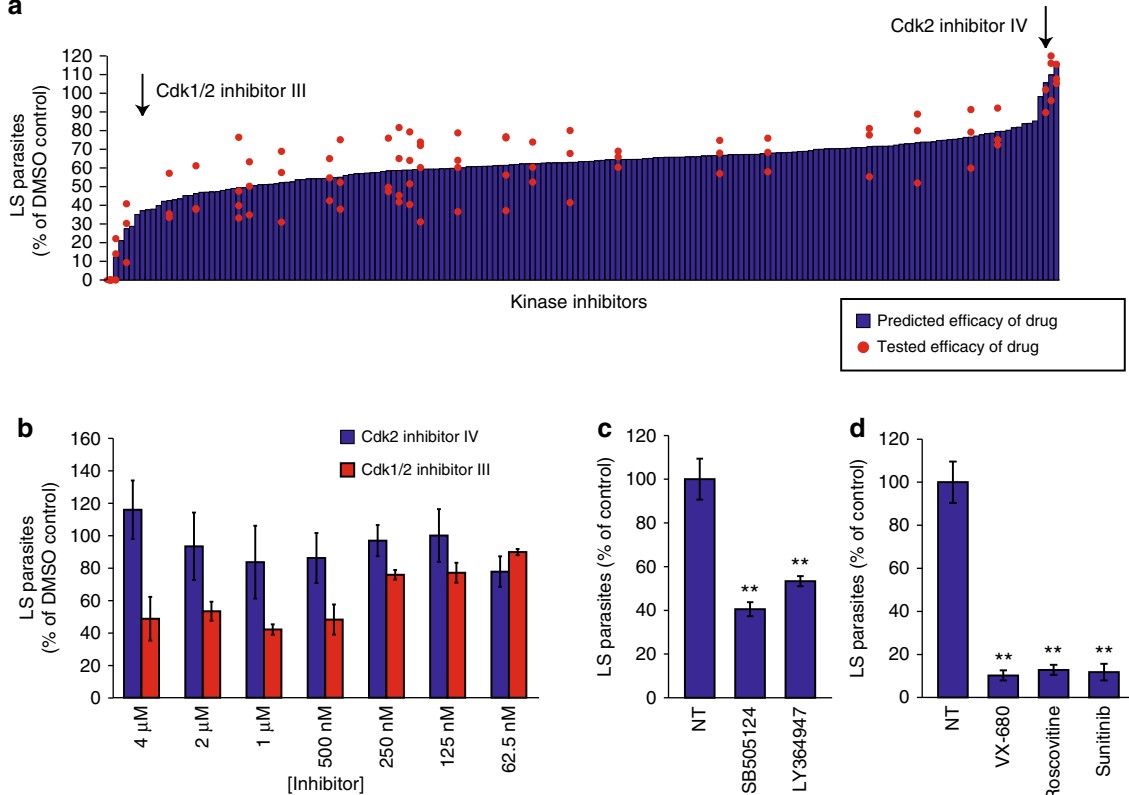

**Fig. 4** Computational prediction of effective host-based drugs against *Plasmodium yoelii* infection. **a** Bar graph depicting predicted efficacy of host-targeted kinase inhibitors in eliminating LS parasite burden. Independent experimental data measuring LS inhibition from drugs used to generate the training dataset are overlaid as data points onto the bar graph. Red arrows indicate predicted efficacies of two drugs against the same kinase target—CDK2. **b** 150,000 Hepa 1–6 cells were infected with 50,000 *P. yoelii* parasites and then treated with CDK2 inhibitors at different concentrations ranging from 4 μM to 62.5 nM at 1.5 h.p.i. LS burden was evaluated by microscopy at 24 h.p.i. **c** 150,000 Hepa 1–6 cells were infected with 50,000 *P. yoelii* parasites and then treated with 500 nM of TGFBR-1 inhibitors SB505124 or LY364947 at 1.5 h.p.i. LS burden was evaluated by microscopy at 48 h.p.i. **p ≤ 0.01 evaluated by Student's two-tailed *t*-test. **d** 150,000 Hepa 1–6 cells were infected with 50,000 *P. yoelii* parasites and then treated with 500 nM VX-680, Roscovitine or Sunitinib at 1.5 h.p.i. Parasite burden was evaluated by microscopy at 24 h.p.i. **p ≤ 0.01 evaluated by Student's two-tailed *t*-test. All data are representative of at least three independent experiments. Error bars represent standard deviation of analytical replicates

**Elastic net regression predicts kinases that regulate LS.** To identify host kinases that regulate *Plasmodium* LS development, we used kinase inhibition data to inform an algorithm based on elastic net regression. This approach is similar to previously described methodology[17]. In short, our methodology regresses experimentally obtained phenotypic data—in our case, parasite burden, against a pre-existing kinase-drug interaction data set[21] (Fig. 2a). Briefly, we modeled the phenotype (residual LS burden), $y$, as a linear function of residual kinase activity X, $y = \beta_0 + X\beta$. Residual kinase activity in response to each of 178 kinase inhibitors has been measured previously using a biochemical kinase activity assay[21]. Parameters of fit ($\beta_0$, $\beta$) were determined using multivariate linear regression with elastic net regularization, which minimizes the error between actual and predicted results. This method has two hyperparameters, $\alpha$ and $\lambda$. $\alpha$ is a relative weighting parameter between 0 and 1 of the elastic net penalty between LASSO regression ($\alpha = 1$) and Ridge regression ($\alpha = 0$) and is chosen to adjust the stringency of inclusion. $\lambda$ is the overall scaling factor of the regularization penalty and is chosen at each value of alpha such that the Mean Squared Error (MSE) of the model, calculated via resubstitution, is minimized (Supplementary Note 1). This approach identified a minimal set of kinases with non-zero coefficients for each value of $\alpha$ (Supplementary Data 1). When $\alpha$ was set at 0.8, 47 kinases had non-zero coefficients (Fig. 2b). We hypothesized that these kinases impact *Plasmodium* LS development.

The predicted set of 47 putative kinase regulators of *Plasmodium yoelii* LS development (Fig. 3a) represents a substantial increase over the current number of kinases known to regulate *Plasmodium* LS infection. To evaluate the efficacy of our approach in identifying kinases with a relevant role in LS infection, we compared predicted kinases to the results of a kinome-wide siRNA screen by Prudêncio and colleagues[24], which identified host kinase regulators of LS infection. Of the 300 kinases that can be evaluated by our approach, seven had been previously identified as regulators of LS infection. Of these seven kinases, four (PKCζ, HCK, c-MET, and MARK2) were predicted by our approach when $\alpha$ was set at 0.8, which is significantly greater than expected by chance ($p = 0.01$, hypergeometric probability test) (Fig. 3a). When $\alpha$ was decreased, relaxing the threshold, two additional kinases (PKCι, SGK2) that had been previously identified[24] were predicted.

Since this methodology provides an unbiased approach to elucidating host regulators of infection, kinases with similar drug-inhibitory profiles have the potential to be predicted as hits together, which could introduce false positives. Thus, to test the accuracy of our predictions, we asked which kinases that we predict to play a role in LS infection significantly impacted LS infection when knocked down using shRNA. We chose to evaluate 18 predicted kinases for their role in LS infection and six non-predicted kinases as negative controls (Fig. 3b, Supplementary Fig. 2). Hepa 1–6 cells were transduced with a pool of shRNA-expressing lentivirus against

each kinase and transduced cells were selected over a period of five days using puromycin. Lentivirus expressing a non-targeting shRNA was used as a control (Supplementary Table 3). Decrease in transcript level of each kinase was evaluated by qPCR (Supplementary Fig. 3, Supplementary Table 4). Each cell line was then infected with *P. yoelii* parasites and parasite development was quantified at 24 h.p.i. by microscopy (Fig. 3b). Knockdown of 15/18 kinases resulted in a substantial decrease in parasite burden, defined as ≤70% of scramble control (estimated false positive rate = 16.7%), despite very minimal cell death after knockdown (Supplementary Fig. 4). Our approach identifies host regulatory kinases with more comprehensive coverage than a whole-kinome siRNA screen, using a less laborious experimental design. Functional classification of kinases identified by our approach revealed enrichment of multiple GO terms and KEGG pathways, including microtubule cytoskeleton reorganization ($p = 0.007$), positive regulation of MAP kinase activity ($p = 0.03$), cytokine-cytokine receptor interaction ($p = 0.03$) and endocytosis ($p = 0.05$) (Fig. 3c) using the DAVID toolbox[25,26]. This suggests that a broad range of cellular activities are likely important for *Plasmodium* LS infection.

**Identification of host-targeted inhibitors of LS infection**. In addition to elucidating host regulatory factors, this approach can be used to predict the efficacy of previously untested host-targeted drugs in eliminating LS burden. The algorithm ranked the 178 kinase inhibitors, for which activity profiles had been obtained[21], for their predicted efficacy against *P. yoelii* LS infection. Inhibitors were ranked from most efficacious (predicted to completely eliminate LS infection) to least efficacious (inhibitors predicted to have no impact or slightly increase LS infection) (Fig. 4a, Supplementary Data 2). Of the 178 compounds, two compounds that were both developed against CDK2 were predicted to have dramatically different efficacies against LS infection. CDK2 inhibitor IV was predicted to have no activity against LS malaria, whereas CDK 1/2 inhibitor III was predicted to have potent efficacy against LS infection. Interestingly, CDK2 was not predicted as a kinase regulator of LS infection, suggesting that the activity of each inhibitor was driven exclusively by 'off-target' effects. To test this prediction, we evaluated LS infection in the presence of each inhibitor over a range of concentrations between 4 μM and 62.5 nM and evaluated LS burden at 24 h.p.i. (Fig. 4b). As predicted, we found that CDK2 inhibitor IV had no significant impact on parasite burden over the range of drug concentrations. In contrast, the predicted efficacious drug, CDK 1/2 inhibitor III, decreased LS burden in a dose-dependent manner. Next, we tested two inhibitors, SB505124 and LY364947, both with high predicted efficacy against the kinase target TGFβ Receptor 1. Both drugs eliminated >40% of LS parasites at 48 h.p.i. when administered at 500 nM ($p = 0.0043$ and $p = 0.0097$, Student's two-tailed t-test) (Fig. 4c). Finally, we evaluated the efficacy of VX-680 and Roscovitine, which are currently in clinical studies as well as Sunitinib, which has been FDA approved. Each of these compounds was predicted to eliminate a substantial portion of LS parasites (Supplementary Data 2). When evaluated experimentally, each of these compounds eliminated >85% of LS burden at 500 nM 24 h.p.i. ($p = 0.0025$, $p = 0.0026$, and $p = 0.0013$ by Student's two-tailed t-test) (Fig. 4d). Taken together, we were able to predict and test host kinases involved in infection and also identify novel host-targeted inhibitors that are effective at dramatically reducing LS infection.

## Discussion

The LS of the *Plasmodium* parasite represents an important interventional opportunity as parasites are only present in small

numbers and clinical symptoms are absent. During this stage, the parasite is entirely dependent on the host hepatocyte, suggesting that the LS parasite might be particularly sensitive to host perturbations. Although it has been understood that this susceptibility could be exploited for intervention, specific strategies for eliminating LS parasites remain limited both in the clinic and in development (reviewed in ref. [27]). No single approach has systematically and broadly identified host factors involved in LS infection and translated these insights to a drug discovery effort. We have demonstrated the ability to identify host kinase regulators of parasite infection within the hepatocyte and directly link those insights to the discovery of host-targeted drugs.

We and others have previously shown that altering the levels of host factors can reduce LS parasite burden[11,13,24,28–33]. Yet, it remains technically challenging to perform a global assessment of the functional role of each human protein in malaria LS infection. Since kinases are known to regulate nearly all outcomes within the cell, evaluating the impact of host kinases indirectly evaluates the impact of a substantially larger portion of the proteome than the kinome itself represents. Previous reports have demonstrated that a number of host kinases are involved in infection[11,24], although inhibition of no single kinase has been demonstrated to completely eliminate infection[24,28]. This could originate from many sources including incomplete knockdown or compensatory activity of other kinases.

To date, the assessment of relevant host factors involved in many cellular processes, including infection against a multitude of pathogens, has relied on genetic screens where pathogen development is observed after knockdown of individual host factors. Previously, Prudêncio and colleagues used a kinome-wide siRNA screen to implicate multiple kinases in the regulation of *Plasmodium* infection and development. Of the kinases they identified to regulate LS infection, seven were included in the 300 kinases that we interrogated in this study. We observed significant overlap between the two approaches. Specifically, four kinases, PCKζ, HCK, c-MET and MARK2, were identified by siRNA to play a role in LS infection, and also predicted by our approach (Fig. 3a). At a relaxed threshold, an additional two kinases are predicted (Fig. 3a).

The substantial agreement of these two approaches is non-obvious for several reasons. First, the systems used to evaluate infection are heterologous: the siRNA screen was performed in Huh7 human hepatoma cells using *P. berghei* infection. In contrast, our study uses Hepa 1–6 mouse hepatoma cells and *P. yoelii* infection. Secondly, the nature of the inhibition between these approaches is inherently different: genetic knockdowns aim to eliminate all aspects of a protein's function (kinase activity in addition to non-kinase related activities) whereas the kinase inhibition screen we have performed only perturbs the catalytic activity of the kinase itself. We have recently demonstrated that at least one member of the Eph family of Receptor Tyrosine Kinases is critical for parasitophorous vacuole membrane formation through engagement of its extracellular (non-kinase) domain[28]. As such, we would expect to see an effect on parasite burden after genetic knockdown but not after kinase domain inhibition. Indeed, Prudêncio et al.[24] identify one member of the Eph receptor family, EphA3, in their genetic screen, but the kinase activity of no members of the Eph family are predicted to be involved in *P. yoelii* LS infection by our approach (Fig. 3a).

Furthermore, we and others have previously demonstrated that host responses and key regulatory factors between *P. yoelii* and *P. berghei* vary[29,34,35]. In our study, we uncovered a surprising difference in the role of MET signaling in LS infection. *P. berghei* relies on MET signaling for infection in all cell types evaluated to date whereas *P. yoelii* does not utilize MET signaling in the HepG2-CD81 model[32]. Interestingly, MET was predicted by our

approach to play a role in *P. yoelii* infection of Hepa 1–6 cells. This suggests that host–parasite interactions can vary across species (and, presumably, different strains of *Plasmodium*) as well as across different host genetic backgrounds. This observation further suggests the need for a straightforward methodology for evaluating host factors involved in infection across many model systems. Ultimately, the capability to interrogate relevant host factors in diverse field-isolated strains, in the context of altered hepatocyte biology, will be critical for our comprehensive understanding of malaria infection in the liver.

Interestingly, PKCζ was previously reported to play a role in LS infection and our approach confirmed its impact. PKCζ is an atypical PKC, which unlike other PKCs, requires neither calcium nor diacylglycerol as second messenger activators. Instead, PKCζ signaling relies on protein scaffolds or other second messengers for activation and interaction with substrates (ref. [36], reviewed in refs. [37,38]). The role of PKCζ in LS infection is consistent with a model where the developing parasite relies exculsively on non-canonical PKC signaling while remaining resistant to changes in canonical PKC signaling, which is activated by a wide variety of cell stimuli. In addition to recapitulating previous findings, we predicted that 43 kinases impact infection that were not identified by previous screens. Among these kinases, several examples of pathways perturbed by the parasite emerge. For example, host cell autophagy has been shown to play a role in LS *Plasmodium* infection as inhibition of key regulators in the autophagic pathway, such as LC3, Beclin, and Atg5 lead to a reduction in parasite size[39]. Consistent with this model, we predict the ULK kinases (ULK1/2/3), which mediate host autophagy, to regulate LS infection.

One potential limitation of this approach is confounding effects that arise from kinase inhibitor activity that is exerted directly on parasite kinases. In order to minimize any direct cidal activities that the kinase inhibitors had on the parasite during LS development, we excluded compounds that inhibited *P. falciparum* parasites during blood stage growth. However, because testing in blood stages is an imperfect surrogate for assessing inhibitor activity against the LS parasite, we performed genetic knockdown experiments to determine the false positive and negative rate of our predictions (Fig. 3b). We validated a subset of the hits and found a false positive rate of 16.7% (3/18). When compared to the previously reported kinome-wide siRNA-screen, we find a false negative rate of 43% (3/7) with the α hyperparameter set at 0.8. This is similar to the rate we observe (50%) when we knockdown individual kinases that were not predicted to play a role in LS infection (Supplementary Fig. 2). Interestingly, several of the kinase knockdowns resulted in a modest effect on LS burden, suggesting that our approach is not comprehensive in its ability to identify kinases that play a role in infection and may exclude the prediction of host kinases that exert a small effect on LS infection. However, this false negative rate is likely an over-estimation as it does not account for kinases whose impact is not tied to their enzymatic activity. As such, the kinases found to be false negatives might have kinase-independent functions. For example, the kinase domain of ABL1 has been described to promote ubiquitination of damaged proteins independently of its kinase activity[40,41]. This activity would not be impaired by chemical inhibition of kinase activity but would be impacted by genetic knockdown.

Of the >500 kinases in the human kinome[42], our approach utilizes activity profiles of 300 kinases. Thus, it is likely that additional kinases play an important role in *Plasmodium* LS development. For example, AMP-activated protein kinase (AMPK) was recently described to play a role in LS infection[43] but was not included in the 300 kinases that we assessed. As additional kinase profiling data becomes available, it will be trivial to re-run our model and identify the contribution of a greater number of host kinases that regulate LS malaria. As new, more selective, and therapeutic-grade kinase inhibitors are profiled for their ability to inhibit a diverse range of host kinases, we will be able to predict their impact on *Plasmodium* development during LS infection using this approach. Finally, as inhibition profiles are generated for inhibitors of other enzyme classes, such as deacetylases, phosphatases and methyltransferases, this method can be extended to make predictions on several key cellular processes not limited to the kinome.

The increased sensitivity of our approach over a forward genetic approach could originate from multiple factors. First, chemical inhibition of kinases is often more complete than siRNA knockdown suggesting that kinases that require a certain threshold of inhibition in order to impact LS malaria may escape detection in an siRNA screen. Secondly, the polypharmacological properties of the kinase inhibitors used in the screen create an environment where multiple kinases are inhibited simultaneously. This enables the identification of groups of kinases whose inhibition is most significantly perturbed in combination. Some of these limitations might be overcome with new genetic approaches using CRISPR/Cas9, as this approach results in more complete or even total knockdown[9,44–46]. However, genetic screens will typically facilitate the knockdown of only a single gene within a single cell. This further emphasizes the unique role that our methodology could play in identifying critical factors involved in a multitude of cellular phenotypes.

The technical advantage of this approach over any other approach previously described to evaluate host factors against LS malaria is substantial. A small-scale, technically straightforward, chemical inhibition experiment can be performed in a wide variety of settings, including using field isolates and evaluating infection in the context of a number of different host genetic backgrounds. As new pathogens emerge and geographic distributions of endemic diseases overlap, it is increasingly critical to be able to rapidly characterize key host points of susceptibility for multiple pathogens as well as in the context of co-infections. By minimizing the resources required to generate data, we can work towards a comprehensive picture of the critical host–pathogen interactions that drive a multitude of infectious diseases worldwide.

## Methods

**Cell lines and culture**. Hepa 1–6 cells were obtained from American Type Culture Collection. Cells were maintained in DMEM-Complete Medium (Dulbecco's modified Eagle medium (Cellgro, Manassas, VA), supplemented with 10% FBS (Sigma-Aldrich, St. Louis, MO), 10,000 IU/ml penicillin/100 mg/ml) streptomycin (Cellgro), 2.5 mg/ml fungizone (HyClone/Thermo Fisher, Waltham, MA and 4 mM L-Glutamine (Cellgro)). Cells were split 2–3 times weekly. All experiments were performed using Hepa 1–6 cells that were passaged between 4 and 12 times after purchase from ATCC.

**Mosquito rearing and sporozoite production**. For *P. yoelii* sporozoite production, female 6-week to 8-week-old Swiss Webster mice (Harlan, Indianapolis, IN) were injected with blood stage *P. yoelii* (17XNL) parasites to begin the growth cycle. Animal handling was conducted according to the Institutional Animal Care and Use Committee-approved protocols. *Anopheles stephensi* mosquitoes were allowed to feed on infected mice after gametocyte exflagellation was observed. Salivary gland sporozoites were isolated according to the standard procedures at 14 or 15 days post-blood meal. For each experiment, sporozoites were extracted from salivary glands in parallel to ensure consistency.

**Quantification of liver stage parasites by microscopy**. $1.5 \times 10^5$ Hepa 1–6 cells were seeded in DMEM-Complete Medium in each well of an eight-well Permanox slide. Cells were infected with $5 \times 10^4$ *P. yoelii* sporozoites per well. Slides were centrifuged for five minutes at $515 \times g$ in a hanging-bucket centrifuge to aid in sporozoite invasion. After 90 minutes, sporozoite-containing media was removed. For treatment/experimental wells, DMEM-Complete Media containing kinase inhibitors at 500 nM was added, or media with 0.5% dimethyl sulfoxide for control wells. Compounds were obtained from Selleck Chemicals (Houston, TX) or MiliporeSigma (Darmstadt, Germany). All compounds were dissolved at 1 mM in dimethyl sulfoxide. Compounds were added to cells 90 minutes after infection.

Parasites were allowed to develop for 24 h or 48 h, at which time cells were fixed with 4% paraformaldehyde, and blocked and permeabilized for 1 h in phosphate-buffered saline (PBS) with the addition of 0.1% Triton X-100 and 2% Bovine serum albumin (BSA). Staining steps were performed in PBS supplemented with 2% BSA. Cells were stained using antisera to *Plasmodium* HSP70 at 4 °C overnight and then washed several times. Antibodies were visualized with the use of AlexaFluor-488 goat anti-mouse secondary antibody (Life Technologies, Grand Island, NY). 4′,6-diamidino-2-phenylindole (DAPI) stain was used to visualize both hepatocyte and parasite nuclei. LS parasites were counted manually by microscopy based on presence of HSP70 fluorescence and confirmed by the presence of parasite nuclei using DAPI staining. Complete well counts were taken for all experiments. Each assay contains data from three to four biological replicates, each which includes three technical replicates. Three individual experiments were performed testing the efficacy of each compound. An additional independent experiment was performed if the standard deviation was ≥20% of the sample mean for the initial three replicates. Parasite numbers in treated wells were normalized to corresponding non-treatment control within experiments in order to assess infection rates; however, non-treatment infection rates were comparable across experiments.

**Plasmodium blood stage assay**. GFP-Luciferase expressing *Plasmodium falciparum* blood stage parasites were cultured asexually. Assays were performed on synchronous (>90%) ring stage parasites at 2% parasitemia and 5% hematocrit. In triplicate, kinase inhibitors were administered to parasites at 500 nM with vehicle as a non-treatment control (0.5% DMSO in PBS), and Chloroquine administered at 2.5 μM as a positive control. At 44 h post treatment, 30 μL of the assay culture was transferred to a 96-well white flat-bottom opaque tissue culture plate (Beckton Dickinson, Franklin Lakes, New Jersey). Using a Berthold LB960 XS3 microplate (Berthold Technologies, Wildbad, Germany) an equal volume of Bright-Glo luciferase reagent (Promega, Madison, WI) was added to parasites and resultant luminescence was measured. Experimental data is representative of three independent experiments.

**Prediction of kinases and kinase inhibitors**. Elastic net regularization, a multivariate variable selection method, was performed using the standard Statistics Toolbox in MATLAB. Specifically, we performed elastic net regularization with a range of alpha values varying from 0.1 to 1.0. Resubstitution methodology was used in order to calculate mean squared error. The code for our analysis is provided in Supplementary Note 1.

**shRNA-mediated gene knockdown**. MISSION shRNA vectors were obtained from Sigma-Aldrich (St. Louis, MO). Detailed information on constructs can be found in Supplementary Table 3. Nonreplicating lentiviral stocks were generated by transfection of HEK293-FT cells. 10 cm TC-treated petri dishes were coated with 0.01 mg/mL poly-L-lysine for ≥30 min at 37 °C, and rinsed with diH2O twice. 4 × $10^6$ HEK293-FT cells were plated on poly-L-lysine coated dishes to achieve 70–80% confluency at time of transfection. Approximately 24 h after plating, transfection mixtures were prepared by mixing 20 μl Polyethylenimine MAX (Polysciences Inc, Warrington, PA) prepared at 1 mg/ml, together with 4.75 μg of shRNA construct or a non-targeting shRNA control, 1.5 μg viral envelope plasmid (pCMV-VSV-G), and 3.75 μg viral packaging plasmid (psPax2). After incubating for 10 minutes at room temperature in DMEM, transfection complexes were added dropwise to cells. After overnight incubation, cells were washed to remove transfection mixtures and were fed with 10 ml fresh media. Lentivirus-containing supernatant was harvested 36 h later, passed through 0.45 μm syringe filters, and either used immediately for transduction or stored at -80 °C.

In order to induce knockdown of candidate host kinases, Hepa 1–6 cells were transduced with lentiviral supernatants in 6-well plates at a cell density of 1 × $10^6$/well. At time of plating, cells were transduced with 1 ml of supernatant in the presence of 1.0 μg/mL polybrene (Sigma-Aldrich St. Louis, MO). In order to select for cells with stable integration of shRNA transgenes, supernatant was replaced with complete media with the addition of 2 μg/mL puromycin 24 h post transduction, and cells were selected for at least five days prior to experiments.

**Validation and quantification of shRNA-mediated knockdown**. Quantification of RNA by real-time RT-PCR: Total RNA was extracted using TRIzol reagent according to the manufacturer's procedure (Invitrogen). cDNA synthesis was performed using the Thermo Scientific ReverAid RT Kit according to the manufacturer's instructions (Thermo Scientific). For this quantitative PCR (qPCR) a standard curve was generated using 1:4 dilutions of a reference cDNA sample for PCR amplification of all target PCR products. The values of each transcript were normalized to mouse GAPDH. Experimental samples were compared to this standard curve to give a relative abundance of transcript.

**Statistical analyses**. Method of statistical analysis are reported for each experiment in the corresponding figure legend.

**Code availability**. Script used to generate kinase and kinase inhibitor predictions is provided as Supplementary Note 1.

**Data availability**. All data generated or analyzed during this study are included in this published article and its supplementary information files or available from the corresponding author upon request.

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

## Acknowledgements

We are grateful to Myles Hollowed for technical assistance. We thank Stefan Kappe for the gift of the *P. falciparum* GFP-Luc strain. We thank Photini Sinnis and Fidel Zavala for the *P. yoelii* Hsp70 antisera. We thank the Center for Infectious Disease Research vivarium staff for their work with mice. All work was done according to IACUC procedures and protocols. This work was funded by 1R01GM101183 (A.K.), 1K99/R00AI111785 (A.K.), K22CA201229 (T.S.G.), and P50CA097186 (T.S.G.).

## Author contributions

N.A., H.S.K., E.K.G., T.B., D.R.D., E.N.F.W. performed experiments and computational analysis. A.K. and T.S.G. supervised the research. A.K. and N.A. wrote the paper with input from all other authors.

## Additional information

**Competing interests:** The authors declare no competing financial interests.

