## [Peer Review File · Nature Communications]

Reviewers' comments:

Reviewer #1 (Remarks to the Author):

Arang and collaborators used 37 established kinase inhibitors to test their ability to eliminate *P. yoelii* infected hepatoma cells. To identify key host kinases for Plasmodium LS development, the authors used the kinase inhibition data to inform an algorithm. Their approach predicted several known as well as a number of novel host kinases regulating liver-stage *P. yoelii*. This is an important study that provides an amazing amount of novel data paving the way for future study to decipher the role of the host cell in the establishment and survival of parasites while developing inside hepatocytes.

I would like though to see a couple of key issues being addressed before the manuscript could be considered for publication:

1. The major concern relies on the possible "contamination" of the results by the parasite kinases. The authors removed from the algorithm the compounds that inhibit the growth of blood stages, considering that these should probably act on the parasite. But the kinases expressed during blood stages are not necessarily the same as those expressed by liver stages, so this "subtraction" may not be adequately realistic. This is further aggravated by the fact that they used *P. falciparum* to perform the blood stage and *P. yoelii* for liver stage trials. While there is not much on what the authors can do experimentally, they should make sure this message is clearly addressed during the discussion.

2. The authors need to carefully revise the manuscript to clearly state that they have identified putative host kinases that regulate *P. yoelii* liver stage infection. While these data provides clues for liver stage infection probably in different Plasmodium spp, none of the data was confirmed in *P. falciparum* for example and as such their conclusion throughout the text, including the abstract, should be carefully revised to take this into consideration.

Other minor points:

1. When the authors state: "We and others have previously demonstrated that reversing a subset of host factors perturbed by Plasmodium infection can reduce or even eliminate LS parasite burden", several references are missing.

Scavenger receptor BI boosts hepatocyte permissiveness to Plasmodium infection. Yalaoui S, et al. Cell Host Microbe. 2008 4(3):283-92.

Host scavenger receptor SR-BI plays a dual role in the establishment of malaria parasite liver infection. Rodrigues CD, et al. Cell Host Microbe. 2008 4(3): 271-82.

GLUT1-mediated glucose uptake plays a crucial role during Plasmodium hepatic infection. Meireles P, et al. Cell Microbiol. 2016 [Epub ahead of print]

2. The expression "reversing a subset of host factors" is ambiguous and scientifically unclear.

Reviewer #2 (Remarks to the Author):

This is an overall well-written and interesting paper that addresses the host cell signaling contribution to the successful establishment of hepatocyte infection by Plasmodium sporozoites. This is of great interest, not only in the context of our fundamental understanding of host-pathogen interactions, but also with respect to the growing importance of host-targeted chemotherapy for infectious diseases.

I have however some concerns:

Major points

1. A major caveat of the study is that, as mentioned in the Introduction, Kinase Regression has been applied so far only to “purely mammalian systems” (line 88), as it is based on inhibition profiles of small-molecule inhibitors on “300 commonly studied kinases”, i.e. mammalian kinases. The facts (i) that the parasite kinome is also present in the present system, and (ii) that many parasite kinases are essential for proliferation and their inhibition would impair EEF development, considerably complicates interpretation of the approach. The avenue proposed by that authors, i.e. to exclude compounds that have a defined threshold effect on erythrocytic stage, goes a long way towards mitigating the problem, but does not solve it, for two reasons: (i) the complement of essential kinases might not be the same in the blood and liver stages; and (ii) the authors propose that the erythrocyte has “limited signaling capacity”; however, host erythrocyte signaling pathway components are clearly involved in infection (PMID 17194200; 17526861) and it has been shown that at least some host erythrocyte kinases are activated by infection (PMID: 21371233). Therefore, the assumption that cidal effects of kinase inhibitors are mediated by parasite kinases in the blood stage experiments may be misguided. In support of the notion that at least some (and probably several) inhibitor do target host erythrocyte kinases is the fact that the compounds excluded by the authors following the blood stage pre-screen are tyrosine kinase inhibitors, and there are no members of the Tyrosine kinase group in the parasite’s kinome. Conversely, some of the inhibitors that were retained have been documented as having an effect on blood stage parasites. For example, staurosporine was shown to inhibit erythrocyte invasion with an IC₅₀ of 250 nM (PMID: 7957765), lower than the 500 nM used here, even though the compound was retained by the authors for the learning algorithm; likewise, Purvalanol A (another compound retained for the analysis) was reported as having a 420nM IC₅₀ on erythrocytic proliferation (PMID: 11434907); there may be other examples. The discrepancy with the present dataset deserves a discussion. Finally, a significant proportion of the inhibitors target kinases involved in cell cycle control that have homologues in the parasite, such as NEKs, GSK-3 or CDKs, and are more likely to be active in the proliferating parasite than in the resting hepatocyte. A contribution from parasite GSK3 or CDKs might explain why some inhibitors of GSK3 (or CDKs, as detailed in Fig. 4), but not others, do come up as hits in the KiR: only those that are able to inhibit the parasite enzyme have an effect... This could easily be tested by in vitro kinase assay using recombinant enzymes (see for example PMID 23214499 for GSK3).

In short, this reviewer is concerned that the parasite kinome may bring a much larger contribution to the phenotypic effects of the inhibitors than discussed in the paper. This may explain the observation (that the authors themselves find surprising, line 166) that “Remarkably, the KiR approach identifies host regulatory kinases with more comprehensive coverage than the existing whole-genome siRNA screen”: there may in fact be false positives due to the contribution of parasite targets, which of course would not affect the experimental siRNA data.

2. It is somewhat surprising that the authors chose to validate their data by siRNA only with seven kinases (out of 44 hits). It would not be that onerous to include the remaining 37 kinases --this would define the false positive rate much better, and significantly enhance the scope of the paper as a source of leads for further drug discovery. It would also be of interest to siRNA-screen a sizeable subset of kinases that are not amongst the 44 hits, to better evaluate the false negative rate as well; a thorough performance evaluation/validation of the approach would be appropriate, especially if one considers the authors’ comment that this approach could be applied to other infection systems. As the authors point out, there may be differences between their system (*P. yoelii*) and the system used in the published kinome screen (*P. berghei*), so the data of a larger siRNA screen would be useful to field.

3. Fig 2: This figure is central to the story but is not easy to read for the non-specialist reader to

understand. The heat map is said to illustrate the residual kinase activity of the 300 kinases treated by the 28 compounds, which is straightforward (it would be useful to include ref 22 in the Figure legend). To facilitate things for the reader, it would be useful to insert the same "inhibitors 1-28" labelling (as shown to the left of the main heat map) to the left of the vertical rectangle labelled "residual LS infection". The legend should define "lambda", and the crucial graph in Fig 2B should be explained more extensively.

Minor / editorial points

Line 23 (Abstract): "...49 kinases, including 5 kinases previously described to impact infection, and 44 novel regulators..." This is not very clear and may seem to imply that the 44 regulators are not kinases. Please rewrite.

Line 53: replace "unfathomable to obtain" with "impossible to obtain".

Lines 59-69. In the context of this paragraph, the paper implicating a host hepatocyte (and erythrocyte) MAPK pathway in *P. berghei* liver stage development (PMID: 21371233) should be cited.

Line 85: straightforward (not "straight forward")

The criteria used for inclusion of inhibitors in the screen are not clear. A statement (line 104) says that 'inhibitors were chosen using the B4 forward selection method". This is obscure to the non-specialist --please briefly explain the principle of selection, and provide a reference if possible.

Line 134: "predicted" instead of "predict"

Reviewer #3 (Remarks to the Author):

The manuscript by Arang et al is well written, describing an innovative approach with interesting and valuable outputs.

The authors utilise a unique approach to attempt to predict the signalling pathways implicated in hepatocyte infection of the malaria parasite using a small training set of kinase inhibitors. The manuscript is concise and provides a good introduction to kinase regression based methodology and the limited understanding of the host factors and regulators that are implicated in liver stage infection. This study provides additional data to previous knockdown approaches in other malaria models with the potential for predictive use in further host-parasite systems.

Minor comments:

Within the methodology section the authors do not provide sufficient details of the experimental conditions, which may be useful to the reader.

Would it be possible to comment on the actual rate of infection with the selected MOI and if this was consistent across biological replicates (not just the normalised data)?

What was the range of the hepatocyte passage across the experimental replicates?

Were all microscopy counts performed manually and was this a complete well count or just field

sampling from the 8 well plates?

Figure 4b. - There should less confidence in the determined EC50 value for Cdk1/2 inhibitor III as only one point is presented on the dose response curve plateau.

Reviewer #4 (Remarks to the Author):

Arang et al developed a computational approach that perform regression analysis on in vitro kinase inhibition profiles of a panel of inhibitors to identify key signaling pathways in liver stage malaria infection. Specifically, they identified a set of Protein Kinase C cascade, which could be validated by a previously published siRNA screen in this system. In addition, the proposed method, Kinase Regression (KiR) recommended some kinases, when tested with inhibitors, showed inhibition in liver stage malaria infection. These kinase inhibitors, VX-680, Roscovitine and Sunitinib, when tested in experiments showed elimination >85% of liver stage burden within 24 hours. The manuscript is well written and organized, and the findings of these inhibitors might be repurposed to treat malaria.

Major Comments:

1. The motivation of KiR is to exploit the polypharmacology aspect of the kinase inhibitors, the authors implemented an elastic net approach in KiR for finding the most informative kinases from the screen for candidate kinases. How is this KiR recommendation compared to the previously published approaches such as KAR (as cited in Refs 18 and 19) or the KISS (Szwajda et al Cell Chemical Biology, 2015, 22, Issue 8, p1144–1155)? Do these methods provide the same / similar list of kinases?
2. Potentially, drug combinations might be better in inhibiting the list of 49 kinases recommended by KiR. Could the authors comment on how to predict combinations of kinase inhibitors to target (majority of) these recommended kinases?
3. Could the authors comments on how to generalize the KiR to other drug-target polypharmacology interactions?

Minor comments:

1. It would be useful to provide the 37 kinase inhibitors data and the raw data that depicted Figure 2A (300 kinases across 28 kinase inhibitors) to reproduce the results.
2. The code for KiR should be made available for the public.

We thank the reviewers for their comprehensive and positive reviews. Below we have provided responses to each of the reviewers' comments.

Reviewer # 1

Arang and collaborators used 37 established kinase inhibitors to test their ability to eliminate *P. yoelii* infected hepatoma cells. To identify key host kinases for Plasmodium LS development, the authors used the kinase inhibition data to inform an algorithm. Their approach predicted several known as well as a number of novel host kinases regulating liver-stage *P. yoelii*. This is an important study that provides an amazing amount of novel data paving the way for future study to decipher the role of the host cell in the establishment and survival of parasites while developing inside hepatocytes.

We thank this reviewer for their kind remarks and positive review.

I would like though to see a couple of key issues being addressed before the manuscript could be considered for publication:

1. The major concern relies on the possible "contamination" of the results by the parasite kinases. The authors removed from the algorithm the compounds that inhibit the growth of blood stages, considering that these should probably act on the parasite. But the kinases expressed during blood stages are not necessarily the same as those expressed by liver stages, so this "subtraction" may not be adequately realistic. This is further aggravated by the fact that they used *P. falciparum* to perform the blood stage and *P. yoelii* for liver stage trials. While there is not much on what the authors can do experimentally, they should make sure this message is clearly addressed during the discussion.

We agree with the reviewer that it is likely we have inadvertently targeted a subset of parasite kinases by some of the compounds used as part of our screen. We also agree that this is, unfortunately, challenging to address experimentally within the drug screen. Off-target effects on parasite kinases, however, cannot occur in the shRNA experiments shown in Figure 3B (below).

Bar graph depicting LS development in cells with shRNA-mediated knockdown of a subset of kinases identified by KiR approach. Values are normalized to non-treated parasites which are indicated by solid line. Threshold for validation is $\leq 70\%$ remaining LS burden after kinase knockdown, indicated by dashed line. KiR predicted kinases with previous reports of LS activity are depicted in red. Novel predicted kinases are depicted in blue. Data is representative of 3 independent experiments. Error bars represent standard deviation of analytical replicates

Despite the potential off-target effects of the kinase inhibitors, our approach is able to accurately predict a wide range of functionally-relevant host kinases in infection. Moreover, we validate many of these predictions via shRNA-mediated knockdown. To clarify this point, we have added the following paragraph to the discussion (line 295):

One potential limitation of this approach is confounding effects that arise from kinase inhibitor activity that is exerted directly on parasite kinases. In order to minimize any direct cidal activities that the kinase inhibitors had on the parasite during LS development, we excluded compounds that

inhibited *P. falciparum* parasites during blood stage growth. However, because testing in blood stages is an imperfect surrogate for assessing inhibitor activity against the LS parasite, we performed genetic knockdown experiments to determine the false positive and negative rate of our predictions (Fig. 3B).

2. The authors need to carefully revise the manuscript to clearly state that they have identified putative host kinases that regulate *P. yoelii* liver stage infection. While these data provides clues for liver stage infection probably in different Plasmodium spp, none of the data was confirmed in *P. falciparum* for example and as such their conclusion throughout the text, including the abstract, should be carefully revised to take this into consideration.

While we agree that although there is likely to be overlap in critical host factors for infection across Plasmodium spp, we cannot make direct conclusions regarding the host signaling requirements of *P. falciparum*. We have revisited the manuscript in detail and specified the use of *P. yoelii* in several places, including the abstract. These additions can be found on lines 23, 25 (the abstract), 85, 105, 109, 154, 194, and 265.

Other minor points:

1. When the authors state: “We and others have previously demonstrated that reversing a subset of host factors perturbed by Plasmodium infection can reduce or even eliminate LS parasite burden”, several references are missing.

Scavenger receptor BI boosts hepatocyte permissiveness to Plasmodium infection. Yalaoui S, et al. Cell Host Microbe. 2008 4(3):283-92.

Host scavenger receptor SR-BI plays a dual role in the establishment of malaria parasite liver infection. Rodrigues CD, et al. Cell Host Microbe. 2008 4(3): 271-82.

GLUT1-mediated glucose uptake plays a crucial role during Plasmodium hepatic infection. Meireles P, et al. Cell Microbiol. 2016 [Epub ahead of print]

We thank this reviewer for their careful review and extensive knowledge of the literature. We have included each of these three references on line 231. These references are now numbers 29, 30 and 31.

2. The expression "reversing a subset of host factors" is ambiguous and scientifically unclear.

We have revised the sentence for clarity as shown below.

We and others have previously shown that altering the levels of host factors can reduce or even eliminate LS parasite burden^{11,13,14,24,26-31}

Reviewer #2

This is an overall well-written and interesting paper that addresses the host cell signaling contribution to the successful establishment of hepatocyte infection by Plasmodium sporozoites. This is of great interest, not only in the context of our fundamental understanding of host-pathogen interactions, but also with respect to the growing importance of host-targeted chemotherapy for infectious diseases.

We thank this reviewer for his/her kind remarks and positive review.

Major points

1. A major caveat of the study is that, as mentioned in the Introduction, Kinase Regression has been applied so far only to "purely mammalian systems" (line 88), as it is based on inhibition profiles of small-molecule inhibitors on "300 commonly studied kinases", i.e. mammalian kinases. The facts (i) that the parasite kinome is also present in the present system, and (ii) that many parasite kinases are essential for proliferation and their inhibition would impair EEF development, considerably complicates interpretation of the approach. The avenue proposed by that authors, i.e. to exclude compounds that have a defined threshold effect on erythrocytic stage, goes a long way towards mitigating the problem, but does not solve it, for two reasons: (i) the complement of essential kinases might not be the same in the blood and liver stages; and (ii) the authors propose that the erythrocyte has "limited signaling capacity"; however, host erythrocyte signaling pathway components are clearly involved in infection (PMID 17194200; 17526861) and it has been shown that at least some host erythrocyte kinases are activated by infection (PMID: 21371233).

We acknowledge that kinase inhibitors have off target effects, including likely, in some cases, on parasite kinases. In some cases, these off target effects might impact the predictions made by the elastic net regression. It is because of this uncertainty, that the KiR approach is designed to make predictions, not conclusions. These predictions are then tested using a genetic approach (shRNA knockdown) that is not susceptible to the same type of off-target effects. This makes the reporting of a false positive rate important. In our revised manuscript, we evaluate an additional set of predicted kinases and as such are able to

estimate a false-positive rate for this approach. This rate is relatively low (16.6%), indicating that the off-target effects likely have at most a modest role in determining KiR predictions. This data is reported in Figure 3B (see below)

One possible reason for the reasonably low contribution from off-target effects is the redundancy that is built into the system. Specifically, multiple kinase inhibitors that are included in our screen have the capacity to inhibit a broad range of host kinases. If one of these inhibitors also has an off-target effect on a parasite kinase, other tested compounds likely do not also target that parasite kinase. As such, the nature of the approach attempts to minimize false positives.

Therefore, the assumption that cidal effects of kinase inhibitors are mediated by parasite kinases in the blood stage experiments may be misguided. In support of the notion that at least some (and probably several) inhibitor do target host erythrocyte kinases is the fact that the compounds excluded by the authors following the blood stage pre-screen are tyrosine kinase inhibitors, and there are no members of the Tyrosine kinase group in the parasite's kinome. Conversely, some of the inhibitors that were retained have been documented as having an effect on blood stage parasites. For example, staurosporine was shown to inhibit erythrocyte invasion with an IC₅₀ of 250 nM (PMID: 7957765), lower than the 500 nM used here, even though the compound was retained by the authors for the learning algorithm; likewise, Purvalanol A (another compound retained for the analysis) was reported as having a 420nM IC₅₀ on erythrocytic proliferation (PMID: 11434907); there may be other examples. The discrepancy with the present dataset deserves a discussion.

We agree with the reviewer that there are likely several examples where a compound eliminates parasite infection in a setting that was described in the literature but not in our hands. It is also possible that different parasite species have slightly or dramatically altered sensitivities to compounds. Specifically, the reviewer cited a reference indicated that *P. knowlesi* is sensitive to staurosporine (PMID: 7957765), but this does not necessarily mean that *P. falciparum* or *P. yoelii* are sensitive. Moreover, it is possible that the precise nature of the blood stage assay performed (Fig. 1B) does not account for toxicity against compounds in all circumstances. The precise nature of the compound also likely contributes to specificity. For example, while, as the reviewers state, Purvalanol A can kill *P. falciparum* parasites (PMID: 11434907), however Aminopurvalanol A, a derivative used in our study does not. However, our intent with this initial pre-screen was to eliminate a portion of the compounds that might contribute to blocking Plasmodium kinases. We acknowledge the reality that our approach does not eliminate possibility of off-target effects. To clarify this point, we have added the following paragraph to the discussion (line 295):

One potential limitation of this approach is confounding effects that arise from kinase inhibitor activity that is exerted directly on parasite kinases. In order to minimize any direct cidal activities that the kinase inhibitors had on the parasite during LS development, we excluded compounds that inhibited *P. falciparum* parasites during blood stage growth. However, because testing in blood stages is an imperfect surrogate for assessing inhibitor activity against the LS parasite, we performed genetic knockdown experiments to determine the false positive and negative rate of our predictions (Fig. 3B).

Finally, a significant proportion of the inhibitors target kinases involved in cell cycle control that have homologues in the parasite, such as NEKs, GSK-3 or CDKs, and are more likely to be active in the proliferating parasite than in the resting hepatocyte. A contribution from parasite GSK3 or CDKs might explain why some inhibitors of GSK3 (or CDKs, as detailed in Fig. 4), but not others, do come up as hits in the KiR: only those that are able to inhibit the parasite enzyme have an effect... This could easily be tested by in vitro kinase assay using recombinant enzymes (see for example PMID 23214499 for GSK3).

We agree that evaluating the effect of compounds on parasite kinases would be reasonably straight forward in instances where parasites kinases could be expressed and purified. However, like the mammalian targets of kinase inhibitors, it is difficult to know precisely what the kinase targets of a given inhibitor are without systematically testing the entire parasite kinome. While this would facilitate an interesting investigation into the activities of parasite kinases, this topic is beyond the scope of the current investigation.

In short, this reviewer is concerned that the parasite kinome may bring a much larger contribution to the phenotypic effects of the inhibitors than discussed in the paper. This may explain the observation (that the authors themselves find surprising, line 166) that “Remarkably, the KiR approach identifies host regulatory kinases with more comprehensive coverage than the existing whole-genome siRNA screen”: there may in fact be false positives due to the contribution of parasite targets, which of course would not affect the experimental siRNA data.

We agree with the reviewer that genetic validation of predictions is critical, as is an estimation of the false positive rate. We have included additional shRNA knockdown experiment in order to further assess the approach. We estimate the false positive rate of 16.6%. These data are included in Fig. 3B (and shown below).

2. It is somewhat surprising that the authors chose to validate their data by siRNA only with seven kinases (out of 44 hits). It would not be that onerous to include the remaining 37 kinases --this would define the false positive rate much better, and significantly enhance the scope of the paper as a source of leads for further drug discovery. It would also be of interest to siRNA-screen a sizeable subset of kinases that are not amongst the 44 hits, to better evaluate the false negative rate as well; a thorough performance evaluation/validation of the approach would be appropriate, especially if one considers the authors' comment that this approach could be applied to other infection systems. As

the authors point out, there may be differences between their system (*P. yoelii*) and the system used in the published kinome screen (*P. berghei*), so the data of a larger siRNA screen would be useful to field.

We agree with the reviewer that a further characterization of the hits by knockdown is beneficial. We have performed additional knockdown experiments in order to more accurately evaluate both the false positive and false negative results of the KiR predictions. These data are reported in Figure 3B (see above) and Supplementary Figure 2. We have also made the following edits to reflect these changes:

We chose 18 predicted kinases to evaluate for their role in LS infection and 6 non-predicted kinases as negative controls (Fig 3B, Supplementary Figure 2).
(line 171)

Knockdown of 15/18 kinases resulted in a substantial decrease in parasite burden, defined as $\leq 70\%$ of scramble control (estimated false positive rate = 16.7%). (line 179)

We validated a subset of the hits and found a false positive rate of 16.7% (3/18). When compared to the previously reported kinome-wide shRNA-screen, we find a false negative rate of 43% (3/7). This is similar to the rate we observe (50%) when we knockdown individual kinases that were not predicted to play a role in LS infection (Supplementary Fig. 2). (line 302)

3. Fig 2: This figure is central to the story but is not easy to read for the non-specialist reader to understand. The heat map is said to illustrate the residual kinase activity of the 300 kinases treated by the 28 compounds, which is straightforward (it would be useful to include ref 22 in the Figure legend). To facilitate things for the reader, it would be useful to insert the same “inhibitors 1-28” labelling (as shown to the left of the main heat map) to the left of the vertical rectangle labelled “residual LS infection”. The legend should define “lambda”, and the crucial graph in Fig 2B should be explained more extensively.

We appreciate the reviewer’s suggestions to increase the readability of this figure for a non-expert. We have inserted the additional labeling in part (A), added the reference to Anastassiadis et al (now reference 22) in the figure legend and further defined details of the methodology in the text and in the figure legend. We did not think that the figure in 2B portrayed clear message that was central to the understanding of the methodology or the results as our computational approach

mirrors what has been reported previously. Hence, we chose to omit the graph in order to facilitate the clarity of the figure. Figure 2 has now been simplified for the non-computational expert and highlights the transformation that occurs from the large dataset (Fig. 2A) to the dataset that only includes predicted kinases (Fig. 2B). Additional technical data, including coefficients of fit at different parameter values can now be found in the supplemental tables.

Additionally, we have defined alpha and lambda in the text and describe our approach in more detail on lines 137-149:

In short, our methodology regresses experimentally obtained phenotypic data—in our case, parasite burden, against a pre-existing kinase-drug interaction dataset²² (Fig. 2A). Briefly, we modeled the phenotype (residual LS burden) y as a linear function of residual kinase activity X , $y = \beta_0 + X\beta$. Residual kinase activity in the response to 178 kinase inhibitors has been measured previously using a biochemical kinase activity assay²². Parameters of fit (β_0 , β) were determined using multivariate linear regression with elastic net regularization, which minimizes the error between actual and predicted results. This method has two hyperparameters, α and λ . α is a relative weighting parameter between 0 and 1 of the penalty regularization between LASSO regression ($\alpha = 1$) and Ridge regression ($\alpha = 0$) and is chosen to adjust the stringency of inclusion. λ is the overall scaling factor of the regularization penalty and is chosen at each value of alpha such that the Mean Squared Error (MSE) of the model, calculated via resubstitution, is minimized.

Minor / editorial points

Line 23 (Abstract): "...49 kinases, including 5 kinases previously described to impact infection, and 44 novel regulators..." This is not very clear and may seem to imply that the 44 regulators are not kinases. Please rewrite.

We have revised the sentence for clarity as shown below:

Our predictions yielded 49 kinases, including five kinases previously described to impact infection, and 44 novel kinase regulators including multiple Receptor Tyrosine Kinases and members of the Protein Kinase C cascade.

Line 53: replace "unfathomable to obtain" with "impossible to obtain".

We have made this replacement in the text.

However, these approaches require large numbers of infected cells which are difficult to generate in laboratory strains and virtually impossible to

obtain when pathogens are obtained from the field or other medically-relevant scenarios.

Lines 59-69. In the context of this paragraph, the paper implicating a host hepatocyte (and erythrocyte) MAPK pathway in *P. berghei* liver stage development (PMID: 21371233) should be cited.

We have the added following sentence which describes these data and included the cited reference:

Inhibition of Mitogen Activated Protein Kinases (MAPKs) in both *Plasmodium*-infected erythrocytes and hepatocytes has also been described to curtail infection¹⁵.

Line 85: straightforward (not “straight forward”)

We have made this replacement in the text.

Kinase Regression (KiR) is an ensemble approach which aims to identify the kinases that most significantly contribute to a biological phenotype¹⁸⁻²⁰ by integrating a straightforward, small-scale kinase inhibitor screen with the tools of computational biology.

The criteria used for inclusion of inhibitors in the screen are not clear. A statement (line 104) says that ‘inhibitors were chosen using the B4 forward selection method’. This is obscure to the non-specialist --please briefly explain the principle of selection, and provide a reference if possible.

Because we based our methods largely on those described in Reference 18, we chose to edit the text to read as follows on line 110:

Thirty-seven kinase inhibitors have previously been described to capture as much of the variability in the inhibitor-kinase activity space as possible using only a modest number of inhibitors¹⁸ (Supplementary Table 1).

Line 134: “predicted” instead of “predict”

We have made this replacement in the text on line 144.

Parameters of fit (β_0 , β) were determined using multivariate linear regression with elastic net regularization which minimizes the error between actual and predicted results.

Reviewer #3

The manuscript by Arang et al is well written, describing an innovative approach with interesting and valuable outputs.

The authors utilize a unique approach to attempt to predict the signaling pathways implicated in hepatocyte infection of the malaria parasite using a small training set of kinase inhibitors. The manuscript is concise and provides a good introduction to kinase regression based methodology and the limited understanding of the host factors and regulators that are implicated in liver stage infection. This study provides additional data to previous knockdown approaches in other malaria models with the potential for predictive use in further host-parasite systems.

We thank this reviewer for their kind remarks and positive review of our manuscript.

Minor comments:

Within the methodology section the authors do not provide sufficient details of the experimental conditions, which may useful to the reader.

Would it be possible to comment on the actual rate of infection with the selected MOI and if this was consistent across biological replicates (not just the normalised data)?

We have added the following line to the methods section to address this comment (line 469):

Parasite numbers in treated wells were normalized to corresponding non-treatment control within experiments in order to assess infection rates; however non-treatment infection rates were comparable across experiments.

What was the range of the hepatocyte passage across the experimental replicates?

We have added the following line to the methods section to address this comment (line 436):

All experiments were performed using Hepa1-6 cells that were passaged between 4 and 12 times after purchase from ATCC.

Were all microscopy counts performed manually and was this a complete well count or just field sampling from the 8 well plates?

We have revised the following line in the methods section to address this comment (line 464)

LS parasites were counted manually by microscopy based on presence of HSP70 fluorescence and validated using DAPI staining. Complete well counts were taken for all experiments.

Figure 4b. - There should less confidence in the determined EC50 value for Cdk1/2 inhibitor III as only one point is presented on the dose response curve plateau.

We appreciate the reviewer's concern. Unfortunately, at the concentrations of Cdk1/2 inhibitor required to achieve maximal inhibition of LS infection there is also hepatocyte death, so we felt it was not appropriate to include this data. Yet, we agree with the reviewer that it is impossible to have substantial confidence in EC50 values calculated this way and instead have opted to show the data at a range of concentrations without reporting an EC50 value. This has led to a revised version of Figure 4B, as shown below:

Reviewer #4:

Arang et al developed a computational approach that perform regression analysis on in vitro kinase inhibition profiles of a panel of inhibitors to identify key signaling pathways in liver stage malaria infection. Specifically, they identified a set of Protein Kinase C cascade, which could be validated by a previously published siRNA screen in this system. In addition, the proposed method, Kinase Regression (KiR) recommended some kinases, when tested with inhibitors, showed inhibition in liver stage malaria infection. These kinase inhibitors, VX-680, Roscovitine and Sunitinib, when tested in experiments showed elimination >85% of liver stage burden within 24 hours. The manuscript is well written and organized, and the findings of these inhibitors might be repurposed to treat malaria.

We thank the reviewer for their kind remarks and positive review.

Major Comments:

1. The motivation of KiR is to exploit the polypharmacology aspect of the kinase inhibitors, the authors implemented an elastic net approach in KiR for finding the most informative kinases from the screen for candidate kinases. How is this KiR recommendation compared to the previously published approaches such as KAR (as cited in Refs 18 and 19) or the KISS (Szwajda et al Cell Chemical Biology, 2015, 22, Issue 8, p1144–1155)? Do these methods provide the same / similar list of kinases?

Target deconvolution using our approach is different from other computational approaches in the following manner:

- 1. *Inhibitor-kinase profiling dataset.*** Our approach utilizes a comprehensive kinase inhibitor profiling dataset (the matrix giving the degree of inhibition for each kinase under the effect of each inhibitor) which is publicly available (Nature Biotechnology 29,1039–1045). This dataset was collected using the ‘gold standard’ radioactive kinase assay to collect high throughput and quantitative information describing the effect of 178 kinase inhibitors on the activity of 300 kinases (using a kinase-specific substrate). For example, since the KAR approach, uses compiled data from multiple sources which are not directly comparable, kinase inhibition data is inputted as a binary term of either inhibited (0) or not inhibited (100).
- 2. *Selection of an “optimal” kinase inhibitor set.*** Another advantage of our approach over the other published approaches is that it uses a representative subset of kinase inhibitors, which explain more than 80% of variance in the kinase inhibitor-kinase space. The ability to make predictions using a minimal set of inhibitors enables us to apply this approach to the context of infectious disease where infectious material is often scarce and large screens are typically intractable.

Although we are not expert at the KAR and KISS approaches, we suspect these approaches could lead to largely overlapping but slightly altered kinase lists. Unfortunately, because of point (2), we did not obtain data that is immediately compatible with using these alternative approaches. As such, we are unable to perform a straight-forward comparison. In further refinements of the approach, it would be interesting to compare different algorithms and then assess the accuracy and overlap of the predictions made by each approach.

2. Potentially, drug combinations might be better in inhibiting the list of 49 kinases recommended by KiR. Could the authors comment on how to predict combinations of kinase inhibitors to target (majority of) these recommended kinases?

Based on 49 ‘informative kinases’ identified using KiR, it is plausible to rationally design combination of kinase inhibitors. Our previous work (PMID: 24707051) has shown that the effect of combination of kinase inhibitors may be predicted with limited success because linear combination of inhibitor effects is too simple assumption. To improve prediction of drug combination there is a need to perform *in vitro* probing of more kinases and at multiple doses. Once this data is available, we will be in excellent position to perform predictions of drug combinations.

3. Could the authors comments on how to generalize the KiR to other drug-target polypharmacology interactions?

Broadly, this approach is also generally applicable to other classes of enzyme inhibitors such as deacetylases, phosphatases and methyltransferases, for which informative target profiles can be obtained. We have added the following text to the discussion to address this point on line 211.

Finally, as inhibition profiles are generated for inhibitors of other enzymes classes such as deacetylases, phosphatases and methyltransferases, this method can be extended to make predictions on several key cellular processes not limited to the kinome.

Minor comments:

1. It would be useful to provide the 37 kinase inhibitors data and the raw data that depicted Figure 2A (300 kinases across 28 kinase inhibitors) to reproduce the results.

We agree with the reviewer that providing this information would be valuable. We have added the LS inhibition data from the 37 kinase inhibitor as Supplementary Table 2. The raw data that was depicted in Figure 2A was taken from a publicly available dataset, cited in reference: 22. We have added the reference in the figure legend for Figure 2A for clarity.

2. The code for KiR should be made available for the public.

We have provided the script as part of the supplementary text:

```
Ts=importdata(' data. csv' );  
Ts. col headers = Ts. textdata(1, 3: end);  
D2T = importdata(' drugs_ data. csv' );  
D2T. ki naseI Ds = D2T. textdata(1, 2: end);  
D2T. drugsI Ds = D2T. textdata(2: end, 1);
```

```

D2T = rmfield(D2T, {'textdata'});
minLambda = 0.05;
maxLambda = 50;
nLam = 100;
Lambdas = minLambda*nthroot(maxLambda/minLambda, nLam - 1).^((1:nLam) - 1);

alphaVals = 0.1:0.1:1;
coeffs = zeros(size(Ts.data, 2) - 1, size(alphaVals, 2));
preds = zeros(size(D2T.data, 1), size(alphaVals, 2));
MSEs = zeros(nLam, size(alphaVals, 2));
constants = zeros(size(alphaVals, 2), 1);
i = 1;
for alpha = alphaVals
    [B, FitInfo] = lasso(Ts.data(:, 2:end), Ts.data(:, 1), 'CV', 'resubstitution',
    'Alpha', alpha, 'MCReps', 1, 'Lambda', Lambdas, 'PredictorNames',
    Ts.textdata(1, 3:end));
    MSEs(1:size(FitInfo.MSE, 2), i) = FitInfo.MSE;
    [~, indx] = min(FitInfo.MSE);
    coeffs(:, i) = B(:, indx);
    B0 = B(:, indx);
    cnst = FitInfo.Intercept(indx);
    constants(i) = cnst;
    B1 = [cnst; B0];
    preds(:, i) = glmval(B1, D2T.data, 'identity');
    i = i+1;
end
alphaLevels = cellstr(strcat({'alpha'}, string(alphaVals)));
alphaLevels = regexp(alphaLevels, '[^a-zA-Z0-9]', '_');
kinNames = regexp(Ts.textdata(1, 3:end), '[^a-zA-Z0-9]', '_');
alphaCoeffs = array2table(coeffs, 'RowNames', kinNames, 'VariableNames',
alphaLevels);
inhNames = regexp(D2T.drugIDs, '[^a-zA-Z0-9]', '_');

```

```

inhPreds = array2table(preds, 'RowNames', inhNames, 'VariableNames',
alphaLevels);
inhPreds.Mean = mean(inhPreds{:, :}, 2);
inhPreds = sortrows(inhPreds, 'Mean');
MSEVals = array2table(MSEs, 'VariableNames', alphaLevels);
MSEVals.Lambda = Lambdas;
MSEVals = [MSEVals(:, size(alphaVals, 2)+1) MSEVals(:, 1: size(alphaVals, 2))];

toc

writetable(inhPreds, 'Inhibitor_Predictions.csv', 'WriteRowNames', true)
writetable(alphaCoeffs, 'Kinase_Predictions.csv', 'WriteRowNames', true)
%writetable(MSEVals, 'MSE.csv')

```

Reviewers' comments:

Reviewer #1 (Remarks to the Author):

The authors have fully addressed my comments and concerns.

Reviewer #2 (Remarks to the Author):

The addition of siRNA experiments is a great improvement to the paper. All kinases whose knock-down results in an impairment in parasite development are clearly involved in parasite development, and the data are compelling --well done (one caveat: for a fully validated dataset, and to formally demonstrate that the effect on LS growth is due directly to interference with this particular kinases rather than secondary effects of dying host cells, it would be nice to show that the viability of cells is not affected by the knock-down). However, my comment on the original submission still holds --now that almost half the hits have been queried by siRNA, it would be straightforward to run the siRNA assay on all 47 hits! This would actually transform the scope of the paper, as it would comprehensively address the question of the identification of potential host kinase targets using this particular approach.

Lines 69-17: "Inhibition of Mitogen Activated Protein Kinases (MAPKs) in both Plasmodium-infected erythrocytes and hepatocytes has also been described to curtail infection 15". The cited paper reports on the effect of inhibition of host erythrocyte MAP kinase kinase (or MEK) (NOT MAP kinase). Please amend. This is important, because in the study in Ref 15, the MAPK is not activated despite the strong activation of MAPKK. Also, please emphasise that the study implicates host erythrocyte/hepatocyte MEK (for which there is no homologue in the parasite's kinome), NOT a parasite-encoded enzyme.

P. 119-125. I question the view that all compounds that are effective on erythrocytic stages should be removed because they target parasite enzymes. There is growing evidence that host erythrocyte kinases are indeed required for parasite proliferation in red blood cells (e.g. Ref 15 and PMID 19131328). This must be mentioned in the manuscript. Indeed, the data provided in the present study provides potentially exciting hypothesis regarding the role of the kinases in Fig 3B in erythrocytic stages as well.

Reviewer #3 (Remarks to the Author):

I believe that the authors have addressed the concerns raised during the initial review in an adequate and appropriate manner.

I have no further comments or concerns.

Reviewer #4 (Remarks to the Author):

The authors have addressed the previous comments, and this revised manuscript is improved for clarity.

We thank all reviewers for their positive review of our work.

Reviewer #2 (Remarks to the Author):

The addition of siRNA experiments is a great improvement to the paper. All kinases whose knock-down results in an impairment in parasite development are clearly involved in parasite development, and the data are compelling --well done (one caveat: for a fully validated dataset, and to formally demonstrate that the effect on LS growth is due directly to interference with this particular kinases rather than secondary effects of dying host cells, it would be nice to show that the viability of cells is not affected by the knock-down). However, my comment on the original submission still holds --now that almost half the hits have been queried by siRNA, it would be straightforward to run the siRNA assay on all 47 hits! This would actually transform the scope of the paper, as it would comprehensively address the question of the identification of potential host kinase targets using this particular approach.

We thank this reviewer for their enthusiastic re-review of the manuscript. We have now included viability data on each of the shRNA knockdown Hepa1-6 cell lines. These data are included in Supplemental Figure 4:

Supplementary Figure 4:

Selective knockdown of hit kinases by lentivirus-mediated shRNA does not induce substantial cell death. Bar graph depicting percentage of trypan-blue positive cells after shRNA knockdown compared to non-targeting control shRNA (scramble). Hepa 1-6 cells were transduced with lentivirus expressing shRNA constructs selectively targeting hit kinases, or a non-targeting control. Percentage of dead cells was assessed by trypan blue staining. Data is representative of two independent experiments. Error bars represent standard deviation of three technical replicates.

Lines 69-17: “Inhibition of Mitogen Activated Protein Kinases (MAPKs) in both *Plasmodium*-infected erythrocytes and hepatocytes has also been described to curtail infection¹⁵”. The cited paper reports on the effect of inhibition of host erythrocyte MAP kinase kinase (or MEK) (NOT MAP kinase). Please amend. This is important, because in the study in Ref 15, the MAPK is not activated despite the strong activation of MAPKK. Also, please emphasize that the study implicates host erythrocyte/hepatocyte MEK (for which there is no homologue in the parasite’s kinome), NOT a parasite-encoded enzyme.

We thank the reviewer for their careful review of our language. We have modified our text to specify that the aspect of component of the MAPK signaling cascade that is activated is MAPKK, not MAPK. We have also emphasized that the MAPKK is of host origin. The sentence on lines 69-71 now reads:

Inhibition of *host* cell Mitogen Activated Protein Kinase Kinase (MAPKKs) in both *Plasmodium*-infected erythrocytes and hepatocytes can also curtail infection¹⁵.

P. 119-125. I question the view that all compounds that are effective on erythrocytic stages should be removed because they target parasite enzymes. There is growing evidence that host erythrocyte kinases are indeed required for parasite proliferation in red blood cells (e.g. Ref 15 and PMID 19131328). This must be mentioned in the manuscript. Indeed, the data provided in the present study provides potentially exciting hypothesis regarding the role of the kinases in Fig 3B in erythrocytic stages as well.

We agree with the reviewer that the role of the kinases that play a critical role in liver stage infection in erythrocytic infection is potentially very exciting! We have re-written the text to clarify this, and have cited the references suggested by the reviewer. The text now reads:

We reasoned that any inhibitor that cleared the parasites during the asexual blood stage might exert its activity by inhibiting *Plasmodium* kinases, and this would confound our modeling approach. Three out of the 37 compounds tested exhibited >25% inhibitory activity against asexual blood stage parasite growth compared to non-treated control. To minimize the confounding effect that the inhibition of parasite kinases these inhibitors were eliminated from further use in our study (Fig. 1A, B). This does not exclude the possibility that erythrocyte kinases play an important role in *Plasmodium* infection, as has been highlighted in other studies^{14,23}.